# The histone chaperone NASP maintains H3-H4 reservoirs in the early Drosophila embryo

**Reyhaneh Tirgar[1], Jonathan P. Davies[1], Lars Plate[1,2], Jared T. Nordman[1]***

1 Department of Biological Sciences, Vanderbilt University, Nashville, Tennessee, United States of America,
2 Department of Chemistry, Vanderbilt University, Nashville, Tennessee, United States of America

* jared.nordman@vanderbilt.edu

**Data Availability Statement:** Proteomic data is freely available through ProteomeXchange under the accession# PXD036809.

**Funding:** This work was supported by National Institutes of Health (NIH) General Medical Sciences

## Abstract

Histones are essential for chromatin packaging, and histone supply must be tightly regulated as excess histones are toxic. To drive the rapid cell cycles of the early embryo, however, excess histones are maternally deposited. Therefore, soluble histones must be buffered by histone chaperones, but the chaperone necessary to stabilize soluble H3-H4 pools in the Drosophila embryo has yet to be identified. Here, we show that CG8223, the Drosophila homolog of NASP, is a H3-H4-specific chaperone in the early embryo. We demonstrate that, while a *NASP* null mutant is viable in Drosophila, *NASP* is a maternal effect gene. Embryos laid by NASP mutant mothers have a reduced rate of hatching and show defects in early embryogenesis. Critically, soluble H3-H4 pools are degraded in embryos laid by NASP mutant mothers. Our work identifies NASP as the critical H3-H4 histone chaperone in the Drosophila embryo.

## Author summary

Histones H2A, H2B, H3, and H4 are small proteins required for the basic unit of DNA packaging. Overexpression or depletion of histones is toxic to cells, therefore histone synthesis, trafficking, storage and deposition into chromatin requires careful regulation. Histone chaperones influence nearly all aspects of histone metabolism, including histone storage. During Drosophila development, excess histones are maternally provided to the early embryo to drive the early rapid nuclear cycles. To prevent histone degradation, maternal deposits of histones must be stabilized by a histone chaperone. To date, however, the histone chaperone that stabilizes H3-H4 in the early embryo has remained unknown. Based on sequence, structure and function, we show that the Drosophila homolog of NASP is the critical H3-H4 histone chaperone in the Drosophila early embryo. NASP is required to stabilize the maternally deposited H3-H4 supply that fuels early embryogenesis. While NASP is not an essential gene, embryos laid by NASP mutant mothers show several defects in embryogenesis. Our work defines fundamental aspects of H3 and H4 storage and stability in the early Drosophila embryo.

awards R35GM133552 to L.P. and R35GM128650 to J.T.N. The funders had no role in study design, data collection and analysis, decision to publish, or preparation of the manuscript.

**Competing interests:** The authors declare that they have no conflict of interest.

## Introduction

Histones are small, highly conserved, and positively charged proteins essential for packaging the eukaryotic genome. The core of chromatin is 147bp of DNA wrapped around an octamer of histones H2A, H2B, H3, and H4 [1–4]. Histone occupancy affects nearly every aspect of chromatin metabolism including transcription, DNA replication, DNA repair and DNA packaging [5–8]. Thus, it is crucial that histone expression levels are delicately balanced as histone reduction or overexpression is detrimental to the cell [9–13]. Exemplifying the importance of histone balance, the production of histones is tightly coordinated with cell cycle progression; histone expression peaks at S phase when the demand for histones is highest [14–17]. Furthermore, the soluble pools of histones are less than 1% of the total histone levels in cells, and mechanisms exist to degrade and prevent the overabundance of soluble histones [16–19].

Early embryogenesis of many organisms, including Drosophila, presents a challenge to the histone supply and demand paradigm. The early Drosophila embryo develops extremely rapidly in the first few hours of development [20,21]. The first 14 nuclear divisions are fast, synchronous, and occur in the absence of zygotic transcription as they alternate between S phase and mitosis in a shared cytoplasm [20]. Therefore, early embryogenesis must be driven from maternally supplied stockpiles of RNA and protein, including histones [22–26]. As blastoderm nuclei enter cycle 10, the cell cycle elongates until nuclear cycle 14, in which the embryo undergoes mid blastula transition (MBT). At this point, maternally deposited RNA is degraded and zygotic transcription ensues [20]. Importantly, soluble histones decrease from 55% in nuclear cycle 11 to less than 1% post-MBT [27]. Thus, there must be mechanisms present in the early embryo to suppress the toxicity associated with excess histones in somatic cells.

From their molecular birth to their eventual deposition into chromatin, histones are continuously bound by a network of proteins known as histone chaperones [28]. Histone chaperones are key for histone stability and affect all aspects of histone metabolism including histone folding, storage, transport, post translational modifications, and histone turnover [29]. Importantly, histone chaperones directly or indirectly affect chromatin structure and function by delivery and handoff of histones to other histone chaperones or chromatin-associated factors within a given network. [30]. While a few chaperones can bind all histones, most histone chaperones bind specifically to H3-H4 or H2A-H2B [29,31–33]. In Drosophila embryos, the histone chaperone Jabba sequesters histones H2A-H2B to lipid droplets and protects H2A and H2B from degradation [34]. It is still unknown, however, what histone chaperone protects soluble H3 and H4 pools in the early embryo. While there are multiple H3-H4-specific histone chaperones, nuclear autoantigenic sperm protein (NASP) is an alluring candidate to chaperone H3-H4 in Drosophila embryos as NASP is known to maintain a soluble reservoir of histone H3-H4 in mammalian cells [25,35]. Furthermore, the Xenopus NASP homolog N1/N2 associates with soluble pools of H3 and H4 in egg lysates [36]. Lastly, *NASP* is essential for embryonic development in mammals, and maternal knockdown of the putative Drosophila NASP homolog led to an arrest in early embryogenesis [37–39]. Thus, we hypothesized that Drosophila NASP is a histone H3-H4 chaperone in the early embryo.

Here, based on sequence, structure and function, we identified CG8223 as the Drosophila NASP homolog. We show that CG8223/NASP specifically binds to histones H3-H4 in vivo. We demonstrate that NASP is a maternal effect gene in Drosophila and that embryos laid by *NASP* mutant mothers have impaired development. Finally, we show that in the absence of NASP, soluble H3 and H4 levels decrease in both eggs and embryos. Overall, our findings demonstrate that NASP protects soluble pools of H3-H4 from degradation in Drosophila embryos.

## Results

### *Drosophila melanogaster* CG8223 is the histone H3-H4 chaperone NASP

In Drosophila, Jabba serves as the major H2A-H2B-specific chaperone, but the H3-H4-specific chaperone has yet to be identified [34]. NASP, Nuclear Autoantigenic Sperm Protein, is an H3-H4-specific chaperone known to buffer excess H3-H4 supply in mammalian cells and Xenopus [35,36]. Previous work has identified CG8223 as a possible NASP homolog based on the conserved Tetratricopeptide (TRP) motifs, which are found in NASP homologs [40]. To verify that CG8223 is in fact NASP, we searched the Drosophila proteome for a homolog of human NASP and identified CG8223 as the one and only putative NASP homolog. Alignment of CG8223 with human NASP revealed a similar domain structure with 28% identity (Panel A of Fig A in S1 Text). Critically, the regions of CG8223 with the highest degree of conservation to human NASP are the regions known to bind to H3 directly (Fig 1A). Furthermore, a structural prediction of CG8223 (excluding the dimerization domain, α-89) is highly similar to a recent human crystal structure of sNASP, with an 1.074 angstrom RMSD value (Fig 1B) [41].

In human cultured cells, NASP is localized to both the nucleus and the cytoplasm [42] or exclusively to the nucleus [43]. To understand CG8223 localization in Drosophila, we stained Drosophila S2 cells with a CG8223-specific antibody. We observed the majority of the signal resides around the periphery of the nucleus (Panel D of Fig A in S1 Text). Given that NASP delivers H3 and H4 for replication-dependent histone deposition, we asked if NASP localization is altered in cells in S phase. Consistent with work in human cells [43], NASP localization was not changed in cells in S phase (Panel E of Fig A in S1 Text).

To test experimentally whether CG8223 is an H3-H4-specific binding protein in vivo, we immunoprecipitated (IP) CG8223 from embryo extracts (0-2h AEL) using a CG8223-specific antibody (Panel F of Fig A in S1 Text). Western blot analysis of the IP revealed that CG8223 and Histone H3.2 (hereby referred to as the replicative Drosophila histone H3) but not H2B, are in the same protein complex (Fig 1C). To extend this analysis beyond H3 and H2B, we used IP coupled to quantitative mass spectrometry to determine which canonical histones and histone variants complex with CG8223. To this end, precipitated material was labelled with tandem mass tag (TMT) and only peptides that were unique to each histone were quantified (Fig 1D). This analysis revealed that CG8223 associates with H3, H4 and H3.3. We did not identify any H3-like centromeric protein Cid peptides in our IPs. Interestingly, we noticed a higher level of H3 and H3.3 in CG8223 IPs relative to H4, suggesting that CG8223 preferentially binds to H3. This is consistent with recent work showing that human NASP has a preference for monomeric H3 [44]. CG8223 does not, however, associate with H2A or H2B (Fig 1E). Lastly, we identified an association between NASP and the H2A variant, H2Av. Based on the conservation, structural similarity and in vivo association with H3-H4, we conclude that CG8223 is the sole Drosophila NASP homolog, which we will now refer to as NASP.

### *NASP* is a maternal effect gene

Now that we have established NASP as a H3-H4-specific binding protein in Drosophila, we wanted to ask how *NASP* affects Drosophila development. We used CRISPR-based mutagenesis to target exon 2 to generate *NASP* mutants. From this approach, we recovered two mutants; *NASP¹* and *NASP²* (Fig 2A). The *NASP¹* allele contains a 6bp insertion resulting in a two amino acid insertion at amino acid 203. Given this small insertion is in a non-conserved region of the protein, it is not predicted to affect NASP function (Panel A of Fig A in S1 Text). In contrast, the *NASP²* allele contains a 4bp deletion that results in a frameshift starting at amino acid 203 and a truncation of NASP (Fig 2A). Western blot analysis of ovary extracts

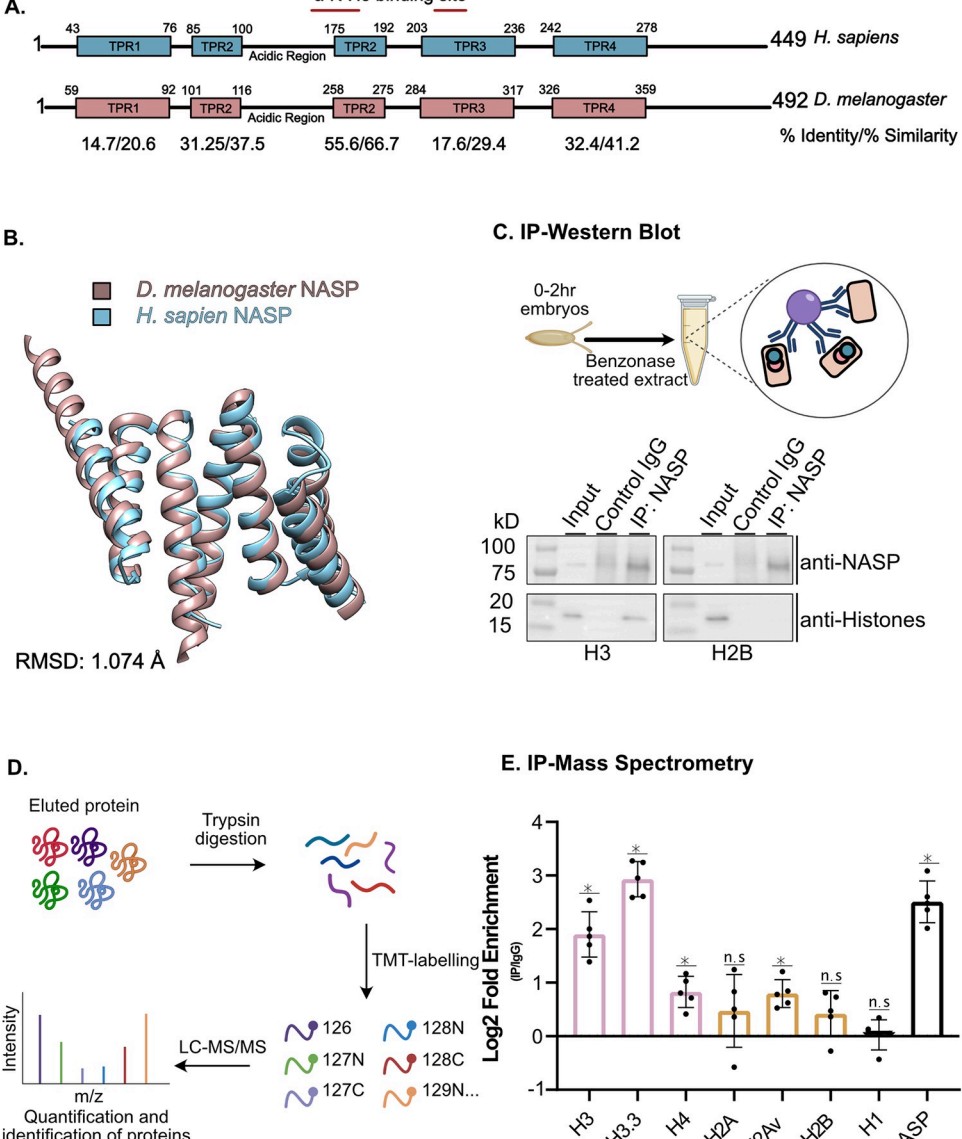

**Fig 1.** ***Drosophila melanogaster* CG8223 is the Histone H3-H4 chaperone NASP. (A)** Schematic of *Homo sapiens* and *Drosophila melanogaster* NASP proteins with TPR domains. Below are the calculated % identity/%similarity for each TPR domains. Red lines indicate the location of residues responsible for binding H3. For specific residues see Panel A of Fig A in S1 text. **(B)** Superimposition of *Homo sapiens* NASP (as determined by crystallography, aa 38–140 210–280) with *Drosophila melanogaster* NASP (predicted by AlphaFold, aa 1–388). **(C)** Immunoprecipitation of NASP from 0-2hr AEL embryos. Methodology created with Biorender. **(D)** Schematic of IP quantitative mass spectrometry approach to quantify NASP-associated proteins created with Biorender. **(E)** Average Log2 fold change for five biological replicates of NASP IP-mass spectrometry relative to IgG control in 0-2hr AEL embryos. Multiple t-test was performed to determine significance (p<0.05).

derived from wild type, *NASP[1]* or *NASP[2]* mutants revealed that there was no detectable NASP[2] protein, even with 4X the protein loaded. In contrast, however, the NASP[1] protein was stable (Fig 2B). To examine viability of the *NASP* mutants, we counted the number of *NASP* mutant progeny relative to the expected frequency (Fig 2C). To account for any CRISPR off target effects, we performed all crosses with two independent deficiency lines (see methods) to

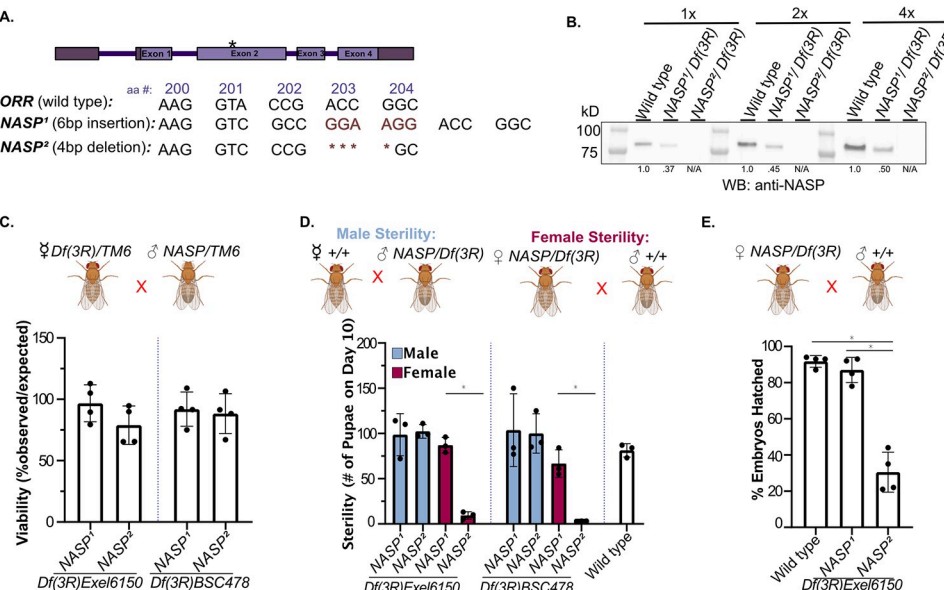

**Fig 2. NASP is a maternal effect gene. (A)** Schematic of *NASP¹* and NASP² CRISPR mutants. **(B)** Western blot analysis of ovary extracts from the indicated genotypes with total protein loading control. *NASP¹/Df(3R)Exel6150* has less NASP due to a reduction in gene dose. Number below lanes represent normalized band quantification relative to wild type. **(C)** The percentage of progeny observed with the appropriate genotype (as shown on the x-axis) over the expected percentage. Each data point is representative of a biological replicate (n = 4). Unpaired t-test was used to determine significance (p<0.05). **(D)** The number of pupae on day 10 produced from females with the genotypes outlined on the x-axis crossed with wild type males. Each data point is representative of a biological replicate (n = 3). Unpaired t-test was used to determine significance (p<0.05). **(E)** Percentage of embryos hatched laid by wild-type, *NASP¹/Df(3R)Exel6150* or *NASP²/Df(3R)Exel6150* mothers. Each data point is representative of a biological replicate (n = 4) and represents the hatch rate of a group of 100 embryos. Dunn's Multiple Comparison post-hoc was performed to determine significance (p<0.05). **(C-E)** Fly crosses created with Biorender.

generate compound heterozygous mutants. Crossing *NASP¹* or *NASP²* mutants with either deficiency line revealed that both *NASP¹* and *NASP²* mutants are viable (Fig 2C).

Although the *NASP²* mutant is viable, it had a lower fecundity (Panel A of Fig B in S1 Text), and we were unable to maintain a stock. Thus, we hypothesized that the *NASP²* mutant is either male or female sterile. To test this hypothesis, we measured the number of pupae formed 10 days after egg laying (AEL) from *NASP* mutant parents. *NASP²* mutant mothers produced a significantly lower number of pupae compared to both wild-type and the *NASP¹* mutant mothers (Fig 2D). There was no significant difference in the number of progeny produced when wild-type females were crossed to male *NASP²* mutants, indicating that loss of NASP function results in female sterility (Fig 2D). Results were consistent for both compound heterozygotes from two independent deficiency lines (Fig 2D).

Previous proteomic studies revealed NASP to be at replication forks in Drosophila cultured S2 cells, Drosophila embryos, and human cells [45,46]. Therefore, it is possible that NASP may function during chorion gene amplification in follicle cells, which is critical to produce egg shell protein in a short developmental window [47]. To test this, we measured DNA copy number at the highest amplified region, *DAFC-66D*, in stage 12 egg chambers. The *NASP²* mutant did not show a significant difference in amplification (Panel B of Fig B in S1 Text). Therefore, we conclude that the female sterility associated with the *NASP²* mutant is independent of gene amplification.

Although *NASP²* mutants were viable, embryos laid by *NASP²* mutant mothers showed a significantly lower hatching percentage compared to *NASP¹* and wild type (Fig 2E). To ask

whether maternally supplied NASP is essential for embryogenesis, we ensured that all progeny have at least one copy of *NASP* by crossing *NASP²* virgin females with wild-type males. Interestingly, even when the progeny had a functional *NASP* allele, there was a significantly lower number of progeny compared to crosses with wild-type females (Panel C of Fig B in S1 Text). Furthermore, embryos laid by *NASP²* mutant mothers crossed with wild type males had a significant reduction in hatching rate (Panel D of Fig B in S1 Text). Therefore, we conclude that *NASP* is a maternal effect gene.

## NASP stabilizes H3-H4 reservoirs in the early Drosophila embryo

Since *NASP* is a maternal effect gene, and embryos laid by *NASP* mutant mothers fail to hatch, embryos laid by NASP mutant mothers are likely devoid of a key factor(s) necessary for development. Given that NASP is a H3-H4-specific chaperone, we hypothesized that H3-H4 reservoirs are destabilized in embryos laid by *NASP* mutant mothers. To specifically measure soluble H3 and H2B reservoirs, we performed Western blot analysis on soluble and total protein extracts in embryos collected from *NASP²* or wild-type mothers (see methods). Qualitatively, embryos laid by *NASP²* mothers had lower levels of soluble and total H3, but not H2B, when compared to embryos laid by wild-type mothers (Fig 3A). To determine when in development H3 pools begin to be degraded in the absence of NASP, we performed Western blot analysis of soluble and total H3 and H2B protein levels in stage 14 egg chambers dissected from *NASP²* and wild-type mothers. In stage 14 egg chambers, soluble, but not total, H3 levels

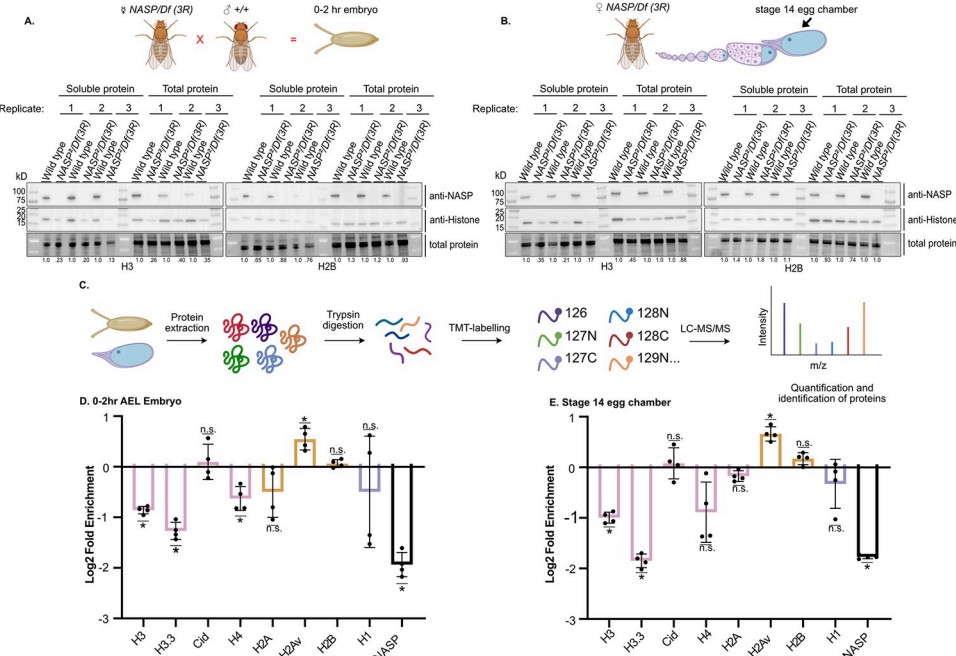

**Fig 3. NASP stabilizes H3/H4 reservoirs in the early Drosophila embryo. (A)** Western blot analysis of soluble and total protein of 0-2hr AEL embryos laid by wild type or *NASP²/Df(3R)Exel6150* mutant mothers. Fly cross created by Biorender. **(B)** Western blot analysis of wild type or *NASP²/Df(3R)Exel6150* stage 14 egg chamber soluble or total protein prepped. Fly schematic created by Biorender. **(C)** Schematic of quantitative mass spectrometry approach to quantify protein abundance created with Biorender. **(D)** Average Log2 fold change for four biological replicates of unique peptides corresponding to H2A, H2Av, H2B, H3, H3.3, H4, Cid, H1, and NASP in 0-2hr AEL embryos laid by *NASP²/Df(3R)Exel6150* or wild-type mothers. Adjusted p-values were calculated by performing multiple t-tests with a Holm-Sidak correction (p<0.05). **(e)** Average Log2 fold change for four biological replicates of unique peptides for H2A, H2Av, H2B, H3, H3.3, H4, Cid, H1, and NASP in *NASP²/Df(3R)Exel6150* or wild type stage 14 egg chambers. Adjusted p-values were calculated by performing multiple t-tests with a Holm-Sidak correction (p<0.05).

were decreased. In contrast, soluble and total H2B protein levels remained the same (Fig 3B). This suggests that in the absence of NASP, H3 forms an insoluble aggregate in stage 14 egg chambers. In support of this, a greater fraction of H3 is found in an insoluble fraction in *NASP²* mutant egg chambers when compared to wild-type egg chambers (Panel A of Fig C in S1 Text). Therefore, in the absence of NASP, H3 is likely prone to aggregation in stage 14 egg chambers but is then degraded in early embryogenesis. Taken together, we conclude that NASP is critical for H3 solubilization and stabilization during both oogenesis and embryogenesis.

To extend this analysis to all canonical and variant histones and gain a quantitative view of histone levels during development, we used quantitative mass spectrometry to measure soluble histone levels in early embryos and stage 14 egg chambers (Fig 3C). To this end, we TMT labeled extracts from 0–2 hour (AEL) embryos and stage 14 egg chambers from four biological replicates. This analysis revealed that the soluble levels of histones H3 and H3.3 were significantly reduced in embryos laid by *NASP²* mutant mothers and in *NASP²* mutant stage 14 egg chambers (Fig 3D and 3E). Soluble H3-like centromeric protein Cid, H4, H1, H2A and H2B levels were stable while H2Av levels increased (Fig 3D and 3E). Overall, quantitative mass spectrometry reveals that in the absence of NASP, soluble pools of histone H3 are reduced starting in oogenesis whereas histone H4 is destabilized in embryogenesis. H3 and H3.3 are more depleted than H4, which is consistent with recent work showing human NASP has a preference for monomeric H3 [44]. Thus, we conclude that NASP stabilizes H3 and H4 soluble reservoirs during both oogenesis and embryogenesis with a preference for H3.

## Embryos laid by *NASP* mutant mothers stall or slow in early embryogenesis

60–70% of embryos laid by *NASP* mutant mothers do not hatch. To determine what the underlying defects are in embryogenesis, we propidium iodide stained 0–4 hour AEL embryos and manually scored the number of embryos in each cell cycle. We observed that 16% of embryos laid by *NASP* mutant mothers were unfertilized, 20% were in cell cycle 1, 31% in cell cycle 2–5, 12% in cell cycles 6–11, and 21% in cell cycles 12 or later. Whereas embryos laid by wild-type mothers were 3% unfertilized, 3% cell cycle 1, 28% cell cycle 6–11, and 57% in cell cycle 12 or later (Fig 4A). This suggests that embryos laid by *NASP* mutant mothers progress more slowly or are stalled in the first embryonic cycles.

To determine if embryos laid by *NASP* mutant mothers are stalled or more slowly progress through the nuclear cell cycles, we collected 0–2 hour AEL embryos and aged them for two hours then scored the embryos for cell cycle stage. The majority of embryos laid by wild type mothers were in cell cycle 12 or later (87%) and only 13% in cell cycles 6–11. In contrast, embryos laid by *NASP* mutant mothers were stalled in an unfertilized stage (20%) or in cell cycle 1 (20%) and only 36% of embryos progressed to cell cycle 12 or later (Fig 4B). Our results are consistent with a recent study showing that maternal depletion of CG8223 results in 61% of embryos arresting in cell cycle stage 2 [39]. To test if embryos laid by *NASP* mutant mothers are associated with DNA damage that could stall or slow the progression of embryogenesis, we measured chromatin bridging. We observed a five-fold increase in chromatin bridging when comparing embryos laid by *NASP* mutant mothers to embryos laid by wild-type mothers (Fig 4C and 4D). Together, we conclude that embryos laid by *NASP* mutant mothers have cell cycle defects that start in the earliest nuclear cycle of embryogenesis.

## Discussion

Early Drosophila embryogenesis provides a unique challenge to histone supply and demand. The early embryo is maternally stockpiled with an overabundance of histones, yet

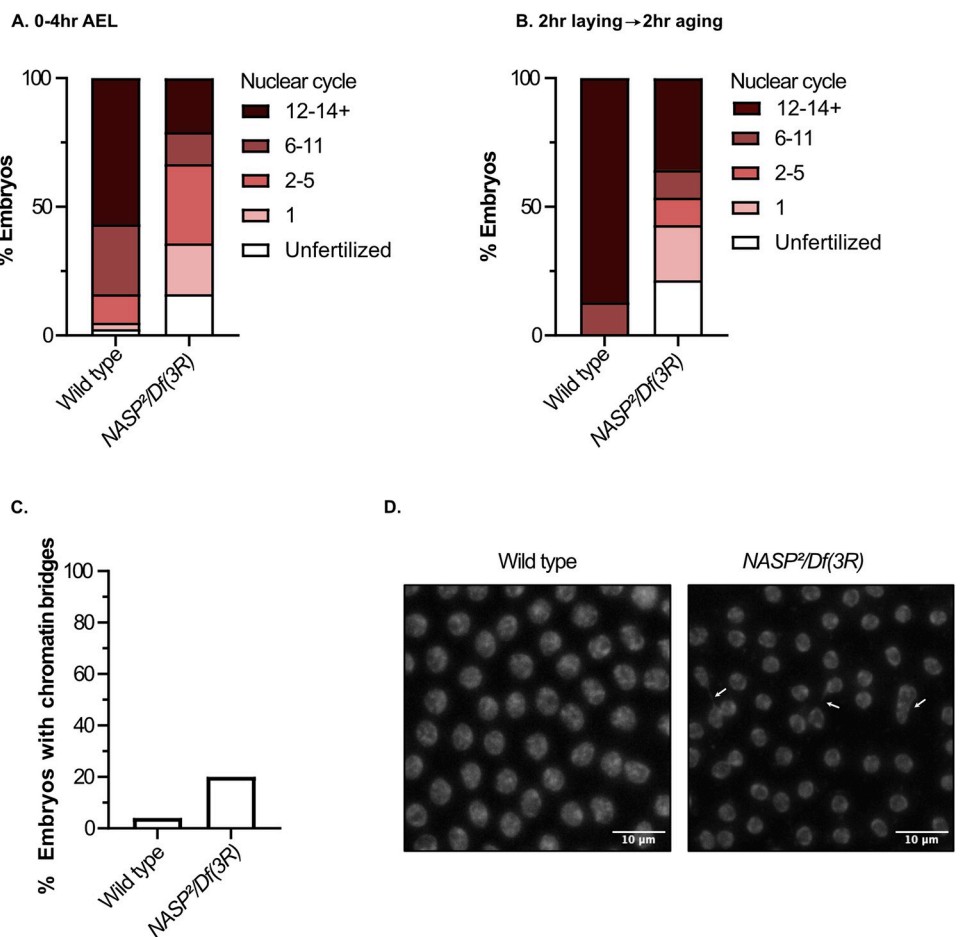

**Fig 4. Embryos laid by *NASP* mutant mothers stall or slow in early embryogenesis. (A)** Percentage of 0-4hr AEL embryos laid by wild type or *NASP²/Df(3R)Exel6150* mutant mothers in respective cell cycles (n = 81). **(B)** Percentage of 0-2hr AEL embryos laid by wild-type or *NASP²/Df(3R)Exel6150* mutant mothers in respective cycles after aging for 2 hours (n = 34).**(C)** Percentage of embryos by wild-type or *NASP²/Df(3R)Exel6150* mutant mothers with chromatin bridging (n = 50 pooled from 3 biological replicates). **(D)**Representative images of propidium iodide stained embryos laid by wild-type or *NASP²/Df(3R)Exel6150* mutant mothers. White arrows point to chromatin bridging.

overproduction of histones is detrimental to cells [10–13,48]. Excess histone supply is likely tolerated through the activity of histone chaperones [25,34]. The histone chaperone Jabba stabilizes soluble H2A-H2B pools in the early embryo by sequestering histones to lipid droplets [34]. The histone chaperone that maintains soluble H3-H4 pools in the early Drosophila embryo, however, has yet to be identified. Our work demonstrates that CG8223, the Drosophila homolog of NASP, is a H3-H4 chaperone in the early embryo. This conclusion is supported by several independent lines of evidence. First, NASP associates with H3-H4, but not H2A-H2B in vivo. Second, soluble pools of H3-H4, but not H2A-H2B, are destabilized in the absence of NASP. Interestingly, NASP binds more H3 than H4 and H3 and H3.3 are depleted more than H4 which is consistent with recent work showing that human NASP preferentially binds H3 [44]. Third, *NASP* is a maternal effect gene and embryos laid by NASP mutant mothers have defects in embryonic development and embryo hatching. Taken together, we conclude that NASP is the predominant H3-H4 chaperone in the early Drosophila embryo.

Embryos laid by *NASP* mutant mothers are stalled in early embryogenesis and display DNA damage. We do not currently know, however, what specific molecular mechanism(s)

underlie these defects. There are several non-mutually exclusive mechanisms that could explain the defects we observe in embryogenesis. First, it is possible that embryos laid by *NASP* mutant mothers are not fertilized. Previously, it has been shown that upon maternal depletion of NASP by RNAi, embryos are still fertilized [39]. It is important to note, however, that these experiments were carried out with a single RNAi line, and the absolute level of depletion was not known. Therefore, it is still possible that embryos laid by *NASP* mutant mothers may not be properly fertilized when NASP is completely eliminated through mutation. Alternatively, it is possible that, similar to embryos laid by *HIRA* mutant mothers, these embryos are fertilized but progress through embryogenesis as haploids [49].

Second, the H3-H4 supply could be insufficient to fuel the demand for chromatin formation in the early embryo. In the early embryo, a single nucleus must rapidly expand to ~8,000 nuclei in two hours [20]. To keep up with the demand for chromatin formation, the early embryo is likely dependent on the maternally loaded histones. In the absence of maternally deposited NASP, H3 and H4 pools are substantially reduced. Thus, it is possible that embryos laid by *NASP* mutant mothers simply lack sufficient H3 and H4 supplies for rapid chromatin formation.

Reduced soluble pools of H3-H4 have the potential to impact cell cycle dynamics in the early embryo. Overexpression of the N-terminal tail of H3 delays Chk1 activation, thereby influencing cell cycle length and the onset of the MBT [50]. Therefore, soluble H3 pools could act as a timer to prevent Chk1 activation and promote rapid cell cycles. Decreasing soluble H3 pools during embryogenesis could allow Chk1 to be prematurely activated and cell cycle length to be extended, thereby altering key cell cycle events in the early embryo and the onset of the MBT [51].

Proper chromatin packaging requires an equimolar ratio of histones [52]. In *C. elegans*, depletion of embryonic H2B levels results in animal sterility. Interestingly, this sterility can be suppressed by reducing H3-H4 levels [53]. Therefore, proper stoichiometry of histones, rather than absolute histone levels, is critical for embryo viability [9,54]. Embryos laid by *NASP* mutant mothers have reduced H3 and H4 levels, yet H2A and H2B levels remain unaffected. It is possible that this alteration in histone stoichiometry leads to both female sterility and defects in embryo development. Similarly, embryos laid by *NASP* mutant mothers have increased H2Av levels. While it is unclear how NASP would directly or indirectly destabilize H2Av, overexpression of H2Av causes nuclear fallout and reduced hatching rate [55]. Thus, the increased H2Av levels in embryos laid by NASP mutant mothers could contribute to defects in embryo development. Finally, while imbalances in soluble histone pools likely contribute to defects during embryogenesis, we cannot rule out the possibility that female sterility in the *NASP* mutant may be caused by changes in the level of a non-histone protein.

It is still unresolved what mechanism is responsible for histone degradation in embryos laid by *NASP* mutant mothers. Our work shows that soluble H3-H4 levels are already reduced in the latest stage of oogenesis, and H3 is likely aggregated in stage 14 egg chambers but not degraded until early embryogenesis. Thus, it will be critical to determine the pathway that degrades unchaperoned H3 and H4 in the early embryo. In mammalian cells, autophagy is responsible for degrading excess H3 and H4 upon NASP depletion [35]. Therefore, it is possible that, in early Drosophila embryos, autophagy is responsible for H3-H4 degradation in the absence of NASP. In embryos laid by *NASP* mutant mothers, we were still able to detect H3 and H4, indicating that H3-H4 pools are not completely degraded in the absence of NASP. This may be because a subset of H3 and possibly H4 are aggregated and not completely degraded. Though it is also possible that another chaperone stabilizes the remaining minor fraction of H3 and H4. For example, HIRA is involved in the deposition of H3.3-H4 dimers in a replication-independent manner while ASF1 and CAF-1 deposit H3-H4 dimers during

replication [56–58]. While these chaperones could stabilize a fraction of the total H3-H4 pools, they are not sufficient to drive embryogenesis. Further, this small fraction of soluble histones could be the result of up regulated translation in the early embryo. In the absence of the H2A-H2B-specific chaperone Jabba, upregulation of translation can compensate for the destabilization of H2A, H2B, and H2Av. It is only when translation is inhibited in the Jabba mutant that embryos die [34]. Now that we have identified NASP as the missing H3-H4-specific chaperone necessary to stabilize soluble H3-H4 pools during Drosophila embryogenesis, we will be able to begin to address fundamental questions about histone storage and stability during oogenesis and embryogenesis.

## Methods

### Strain list

Wild type–Oregon R (OrR)

NASP null mutant (NASP$^2$)- w[1118]; Df(3R)Exel6150, P{w[+mC] = XP-U}Exel6150/TM6B, Tb [1]/NASP$^2$ or w[1118]; Df(3R)BSC478/TM6C, Sb [1] c u[1]/NASP$^2$

NASP control mutant (NASP$^1$)- w[1118]; Df(3R)Exel6150, P{w[+mC] = XP-U}Exel6150/ TM6B, Tb [1]/NASP$^1$ or w[1118]; Df(3R)BSC478/TM6C, Sb [1] cu [1]/NASP$^1$

### CRISPR mutagenesis

To generate a null allele of NASP, a single gRNA targeting exon 2 of the CG8223 was cloned into pU6-BbsI plasmid as described [59]. The gRNA was identified using the DRSC Find CRISPRs tool (http://www.flyrnai.org/crispr2/index.html). The gRNA-expressing plasmid was injected into a nos-Cas9 expression stock (Best Gene Inc.). Surviving adults were individually crossed to TM3/TM6 balancer stock and progeny were screened by Sanger sequencing. The NASP$^1$ allele contains a 6bp insertion resulting in a two amino acid insertion at amino acid 203. NASP$^2$ allele contains a 4bp deletion that results in a frameshift starting at amino acid 203 and a premature truncation of NASP.

### Antibodies and antibody production

The NASP ORF was cloned into the 6His-MBP-containing expression vector pLM302 (Vanderbilt Center for Structural Biology). 6His-MBP- tagged NASP was expressed in E. coli Rossetta DE3 cells (Millipore Sigma, Cat# 71400–3) and purified using MBP Agarose beads (Qiagen). The purified protein was used for injection (Cocalico Biologicals Inc.). NASP antiserum was produced in rabbits. Rabbit anti-NASP antibody was used for western blot (1,2000) and immunoprecipitation.

### Protein alignment and structural prediction

Protein sequence alignments were performed with MAFFT (default settings) and visualized on Jalview. Sequence identities and similarities were generated on SIAS (http://imed.med.ucm.es/ Tools/sias.html) with default settings.

The structure of human NASP was previously solved by X-ray crystallography [41].The structure of Drosophila NASP was predicted using the AlphaFold Protein Structure Database (Q9I7K6). The α-89 was manually removed from the PDB files using PDBTOOLS [60–62]. Superimposition and RSD values of Human NASP core crystal structure (aa 38–140 210–280) and Drosophila NASP predicted structure(aa 1–388) were generated with USCF Chimera. Superimposition was performed on Matchmaker with default settings.

## Viability, sterility, and fecundity assays

For viability assays, *NASP¹* or *NASP²* virgin females were crossed with male *Df(3R)* flies. The genotype of adult progeny was identified using visible markers. The percentage of viability was calculated as (#observed/# expected) *100. For sterility assays, *NASP¹/ Df(3R)* or *NASP²/Df (3R)* females or males were incubated with *OrR* males or *OrR* virgin females, respectively. After three days, adult flies were removed and the number of pupae were scored on day ten. As a control, *OrR* females were crossed to *OrR* males. For Fecundity assays, seventy *NASP²/Df (3R) or OrR* female flies were incubated with *OrR* male flies in a bottle capped by a grape juice agar plate with wet yeast for embryo collection. Collection plates were changed twice in one-hour increments prior to collections. 0–2-hour (AEL) embryos were collected and scored.

## Embryo hatching assay

Embryos laid by *NASP¹/ Df(3R)* or *NASP²/Df(3R)* mothers were collected on grape juice agar plates with wet yeast. One hundred 0–24-hour after egg laying (AEL) unhatched embryos were transferred to a fresh grape juice plate and incubated at 25˚C overnight. Unhatched embryos were scored after 24 hours of incubation. Four hundred embryos were scored for each genotype.

## Copy number profiling

Ovaries were dissected from *NASP¹/ Df(3R)*, *NASP²/Df(3R)* or *OrR* females fattened for two days on wet yeast in Ephrussi Beadle Ringers (EBR). Stage 12 egg chambers were isolated, re-suspended in LB3 [63] and sonicated using a Bioruptor 300 (Diagenode) for five cycles of 30s on and 30s off at maximal power. Lysates were treated with RNase and Proteinase K and genomic DNA was isolated via phenol-chloroform extraction. qPCR was performed using primers previously described [64].

## Cytology and microscopy

*NASP²/Df(3R) or OrR* female flies were incubated with *OrR* male flies in a bottle capped by a grape juice agar plate with wet yeast for embryo collection. Collection plates were changed twice in one-hour increments prior to collections. For staging experiments, 0–4 hour (AEL) embryos were collected. For aging experiments, 0–2 hour (AEL) embryos were collected then aged for two hours. Both samples were dechorionated by 50% bleach for two minutes. Embryos were thoroughly washed with water then dried for 30s. Embryos were transferred to a scintillation vial containing 1mL of heptane. An equal volume of methanol was added and the vial was vigorously shaken by hand for two minutes. Embryos were allowed to settle; the heptane layer was removed and embryos were quickly rinsed with methanol thrice. Embryos were kept in methanol at 4˚C until staining. Once ready for staining, embryos were gradually rehydrated in increasing concentration of PBS (18.6mM $NaH_2PO_4$, 84.1mM $Na_2HPO_4$, 1.75M NaCl,pH 7.4). Embryos were then rinsed in PBX (18.6mM $NaH_2PO_4$, 84.1mM $Na_2HPO_4$, 1.75M NaCl, 0.1% Triton X-100, pH 7.4) for five minutes on a nutator. Then, embryos were treated with 0.8mg/mL RNase A (Macherey-Nagel, 740505) for one hour at 37˚C. After incubation, embryos were washed with PBX for 30 minutes then stained with Propidium Iodide (0.1μg/mL) in PBS for 15 minutes at room temperature. After staining, embryos were washed with PBX for one hour and mounted with VECTASHIELD mounting medium (Vector Laboratories, H-1200). Images were taken at 40X on a Nikon Ti-E inverted microscope with a Zyla sCMOS digital camera. Embryos were manually staged [65] (Panel A of Fig D in S1 Text) and

scored for chromatin bridging. Representative images were rendered with maximum projection intensity.

S2 cells were obtained from the Drosophila Genomics Resource Center (DGRC) and were confirmed negative for mycoplasma contamination via MycoStrip (Invivogen,rep-mys-10). Cells were grown in Schneider's Drosophila Medium (Thermo Fisher Scientific, 21720001) supplemented with 10% heat-inactivated FBS (Gemini Bio Products, 900–108) and 100 U/mL of Penicillin/Streptomycin (Thermo Fisher Scientific, 15070063) at 25˚C.

For NASP localization, Cells were washed with PBS then attached to Concanvan A-coated slides for two hours. Once attached, cells were fixed for 15 minutes in 4% paraformaldehyde and permeabilized with PBX for 15 minutes. Cells were blocked with blocking buffer (PBX supplemented with 1% BSA and 2% goat serum) for one hour then incubated with the NASP primary antibody at a 1:1000 dilution overnight at 4˚C. After overnight incubation, cells were washed with PBX thrice five minutes each followed with an extensive two hour incubation with the secondary antibody (Life technologies, A11011) at a 1:500 dilution. Cell were then washed with PBX thrice five minutes each. Cells were stained with DAPI (1μg/mL) in PBS for 15 minutes at room temperature then washed with PBS for 10 minutes before being mounted with VECTASHIELD mounting medium (Vector Laboratories, H-1200). Images were taken at 40X on a Nikon Ti-E inverted microscope with a Zyla sCMOS digital camera. Profile intensities were generated on the NIS Elements AR 3.2 software.

For S phase analysis, S2 cells were pulsed with 20 μM of CldU nucleoside (Sigma-Aldrich, C6891) for 20 minutes. Cells were washed with PBS then attached to Concanvan A-coated slides for two hours. Once attached, cells were fixed for 15 minutes in 4% paraformaldehyde and permeabilized with PBX for 15 minutes. Cells were acid treated with 2N HCl for 30 minutes then neutralized for two minutes in 0.1M Sodium Borate. Cells were washed with PBX three times for 10 minutes each then blocked with blocking buffer (PBX supplemented with 5% goat serum(Sigma-Aldrich, G9023-10ML)) for 30 minutes at room temperature. After blocking, cells were incubated with the primary antibodies (NASP 1:1000, CldU 1:25 Abcam ab6326) in blocking buffer overnight at 4˚C. After overnight incubation, cells were washed with PBX three times for five minutes each. They were once again blocked for 30 minutes at room temperature then probed with secondary antibodies for two hours at room temperature. For NASP, secondary antibodies Alexa flour 488 Goat anti-Rabbit (Life technologies, A11034) was used at 1:500. For CldU, Goat anti-Rat 594 (Abcam, ab15160) was used at 1:350. Cells were rinsed three times with PBX then washed four times five minutes each. Cells were stained with DAPI (1μg/mL) in PBS for 15 minutes at room temperature then washed with PBS for 10 minutes before being mounted with VECTASHIELD mounting medium (Vector Laboratories, H-1200). Images were taken at 40X on a Nikon Ti-E inverted microscope with a Zyla sCMOS digital camera. Profile intensities were generated on the NIS Elements AR 3.2 software.

## Tissue collection and western blotting

*NASP[1]/ Df(3R) or NASP[2]/Df(3R)* female flies were fattened on wet yeast for 3–4 days, ovaries were dissected in EBR and stage 14 egg chambers were isolated. For embryo isolation, 0–2 hour (AEL) embryos were collected from *NASP[1]/ Df(3R) or NASP[2]/Df(3R)* mothers as described above. For total protein preparation, embryos and egg chambers were homogenized with a pestle in 2x Lammeli buffer (Bio-Rad, 1610737) supplemented with 50mM DTT, boiled for five minutes and loaded onto a Mini-PROTEAN TGX Stain-Free Gel (Bio-Rad). For soluble protein preparations, embryos and egg chambers were flash frozen, and stored at –80˚C until use. Samples were thawed on ice and 50 μL of NP40 lysis buffer was added. Samples were

then homogenized five times with a B-type pestle, transferred into a 1.5mL Eppendorf tube and centrifuged for 30s at 10,000RCF at 4°C. 50 µL of supernatant was transferred to a new 1.5mL Eppendorf tube for protein precipitation. Samples were precipitated using methanol: chloroform:water (3:1:3) and washed three times with methanol. Each wash was followed by a five minute spin at 10,000xg at room temperature. Protein pellets were air dried and resuspended in 10µL of 1% Rapigest SF then denatured with an equal volume of Lammeli buffer supplemented with 50mM DTT. Samples were boiled for five minutes and loaded onto a Mini-PROTEAN TGX Stain-Free Gel.

For the fractionation assay, egg chambers were flash frozen, and stored at –80°C until use. Samples were thawed on ice and 50 µL of NP40 lysis buffer was added. Samples were then homogenized five times with a B-type pestle, transferred into a 1.5mL Eppendorf tube. Input was aliquoted then samples were centrifuged for 30s at 10,000RCF at 4°C. Supernatant was aliquoted into a new 1.5mL Eppendorf tube then the pellet was re-suspended with equal volume NP40 as supernatant. An equal volume of 2x Lammeli buffer (Bio-Rad, 1610737) supplemented with 50mM DTT was added to the input, pellet, and supernatant samples. Samples were boiled for five minutes and loaded onto a Mini-PROTEAN TGX Stain-Free Gel (Bio-Rad). After electrophoresis, the gel was activated and imaged using a BioRad ChemiDoc MP Imaging System following manufacturer recommendations. Protein was transferred to a low fluorescence PVDF membrane using a Trans-Blot Turbo Transfer System (Bio-Rad). Membranes were blocked with 5% milk in TBS-T (140mM NaCl, 2.5mM KCl, 50 mM Tris HCl pH 7.4, 0.1% Tween-20) for 10 minutes. Blots were incubated with the primary antibody (anti-NASP-1:2000, anti-H3-1:000, anti-H2B-1:1000) for one hour at room temperature. Blots were washed three times with TBS-T then incubated with the secondary antibody (HRP anti-mouse-1:20,000, HRP anti-Rabbit-1:25,000) for 30 minutes at room temperature. After hybridization, blots were washed three times with TBS-T then incubated with Clarity ECL solution (Bio-Rad) before imaging. All blots were imaged on the BioRad ChemiDoc MP Imaging System.

### Immunoprecipitation and western blotting

Embryos from *OrR* flies were collected from a population cage. Plates were cleared for one hour, then 0–2 hour embryos (AEL) (pre-MBT) were collected, dechorionated in 50% bleach and flash frozen in nitrogen. Embryo staging was confirmed by DAPI staining (Panel C of Fig A in S1 Text). Embryos were disrupted by grinding them with a mortar and pestle in liquid nitrogen. The powdered embryos were thawed and resuspended on ice in NP40 lysis buffer (50mM Tris-Cl pH 7.4, 150mM NaCl, 1% NP40, 1mM EDTA, 1mM EGTA) supplemented with 2X cOmplete Protease Inhibitor Cocktail EDTA-free (Millipore Sigma). Once thawed, the extract was treated with benzonase at a final concentration of 30 U/ml (EMD Millipore, 70664-10KUN) for 30 minutes on ice. After benzonase treatment, extract was centrifuged at 4000xg for five minutes. Supernatant was used as the starting material for immunoprecipitations. Rabbit IgG (negative control) or NASP serum were added to lysates and incubated at 4°C for two hours. After antibody incubation, prewashed Protein A Dynabeads (Thermo Fisher Scientific, 10001D) were added to the extract and incubated for one hour at 4°C on a nutator. After incubation, beads were isolated and washed once with NP40 lysis buffer, twice with NP40 lysis high salt wash buffer (50mM Tris-Cl pH 7.4, 500mM NaCl, 1% NP40, 1mM EDTA, 1mM EGTA), and once again with NP40 lysis buffer. Beads were then resuspended in 2x Laemmli sample buffer (Bio-Rad, 1610737) supplemented with 50mM DTT and boiled for five minutes to elute protein. Western blot analysis was performed as described previously (Tissue collection and western blotting).

## Mass spectrometry sample preparation

For NASP-immunoprecipitation (IP), samples were prepared as described previously (Immunoprecipitation and Western blotting). For soluble protein levels in stage 14 egg chambers and embryos, 20 embryos or stage 14 egg chambers were collected for each replicate, flash frozen, and stored at –80˚C until use. Once all samples for four biological replicates were collected, samples were thawed on ice and a 100 μL of NP40 lysis buffer was added. Samples were then homogenized ten times with a B-type pestle, transferred into a 1.5mL Eppendorf tube and centrifuged for 30s at 10,000RCF at 4˚C. 100 μL of supernatant was transferred to a new 1.5mL Eppendorf tube for protein precipitation.

Both lysate and IP samples were precipitated using mass spectrometry grade methanol: chloroform:water (3:1:3) and washed three times with methanol. Each wash was followed by a five minute spin at 10,000xg at room temperature. Protein pellets were air dried and resuspended in 5μL of 1% Rapigest SF. Resuspended proteins were diluted with 32.5 μL mass spectrometry grade water and 10 μL 0.5 M HEPES (pH 8.0), then reduced with 0.5 μL of 0.5 M TCEP (freshly made) for 30 minutes at room temperature. Free sulfhydryl groups were acetylated with 1 μL of fresh 0.5 M Iodoacetamide for 30 minutes at room temperature in the dark and digested with 0.5 μg trypsin/Lys-C (Thermo Fisher) overnight at 37˚C shaking. Digested peptides were diluted to 60 μL with water and labeled for 1 hour at room temperature using 16plex TMTpro (Thermo Scientific) or 10plex TMT (Thermo Scientific) for lysate and IP samples, respectively. Labeling was quenched with the addition of fresh ammonium bicarbonate (0.4% v/v final) for one hour at room temperature. Samples were pooled, acidified to pH < 2.0 using formic acid, concentrated to 1/6th original volume via Speed-vac, and diluted back to the original volume with buffer A (95% water, 5% acetonitrile, 0.1% formic acid). Cleaved Rapigest products were removed by centrifugation at 17,000xg for 30 minutes and supernatant transferred to fresh tubes for storage at -80˚C until mass spectrometry analysis.

## MudPIT liquid chromatography-tandem mass spectrometry

Triphasic MudPIT columns were prepared as previously described using alternating layers of 1.5cm C18 resin, 1.5cm SCX resin, and 1.5cm C18 resin [66]. Pooled TMT samples (roughly one-third of pooled IP samples and roughly 20 μg of peptide from lysate samples) were loaded onto the microcapillaries using a high-pressure chamber, followed by a 30 minute wash in buffer A (95% water, 5% acetonitrile, 0.1% formic acid). Peptides were fractionated online by liquid chromatography using an Ultimate 3000 nanoLC system and subsequently analyzed using an Exploris480 mass spectrometer (Thermo Fisher). The MudPIT columns were installed on the LC column switching valve and followed by a 20cm fused silica microcapillary column filled with Aqua C18, 3μm, C18 resin (Phenomenex) ending in a laser-pulled tip. Prior to use, columns were washed in the same way as the MudPIT capillaries. MudPIT runs were carried out by 10μL sequential injections of 0, 10, 20, 40, 60, 80, 100% buffer C (500mM ammonium acetate, 94.9% water, 5% acetonitrile, 0.1% formic acid) for IP samples and 0, 10, 20, 30, 40, 50, 60, 70, 80, 90, 100% buffer C for global lysate samples, followed by a final injection of 90% C, 10% buffer B (99.9% acetonitrile, 0.1% formic acid v/v). Each injection was followed by a 130 min gradient using a flow rate of 500nL/min (0–6 min: 2% buffer B, 8 min: 5% B, 100 min: 35% B, 105min: 65% B, 106–113 min: 85% B, 113–130 min: 2% B). ESI was performed directly from the tip of the microcapillary column using a spray voltage of 2.2 kV, an ion transfer tube temperature of 275˚C and a RF Lens of 40%. MS1 spectra were collected using a scan range of 400–1600 m/z, 120k resolution, AGC target of 300%, and automatic injection times. Data-dependent MS2 spectra were obtained using a monoisotopic peak selection mode: peptide, including charge state 2–7, TopSpeed method (3s cycle time), isolation

window 0.4 m/z, HCD fragmentation using a normalized collision energy of 36% (TMTpro) or 32% (TMT 10plex), resolution 45k, AGC target of 200%, automatic (lysate) or 150 ms (IP) maximum injection times, and a dynamic exclusion (20 ppm window) set to 60s.

## Peptide identification and quantification

Identification and quantification of peptides were performed in Proteome Discoverer 2.4 (Thermo Fisher) using a UniProt *Drosophila melanogaster* proteome database (downloaded February 6[th], 2019) containing 21,114 protein entries. The database was adjusted to remove splice-isoforms and redundant proteins and supplemented with common MS contaminants. Searches were conducted with Sequest HT using the following parameters: trypsin cleavage (maximum 2 missed cleavages), minimum peptide length 6 AAs, precursor mass tolerance 20ppm, fragment mass tolerance 0.02 Da, dynamic modifications of Met oxidation (+15.995 Da), protein N-terminal Met loss (-131.040 Da), and protein N-terminal acetylation (+42.011 Da), static modifications of TMTpro (+304.207 Da) or TMT 10plex (+229.163 Da) at Lys and N-termini and Cys carbamidomethylation (+57.021 Da). Peptide IDs were filtered using Percolator with an FDR target of 0.01. Proteins were filtered based on a 0.01 FDR, and protein groups were created according to a strict parsimony principle. TMT reporter ions were quantified considering unique and razor peptides, excluding peptides with co-isolation interference greater that 25%. Peptide abundances were normalized based on total peptide amounts in each channel, assuming similar levels of background in the IPs. Protein quantification used all quantified peptides. Post-search filtering was done to include only proteins with two identified peptides. Unique peptides for each canonical and variant histone was manually identified, summed, and statistically analyzed on Graphpad Prism. For IP samples, multiple t-test was performed ($p < 0.05$). For lysate samples, multiple t-test with Holm-Sidak correction was performed ($p < 0.05$).

## Supporting information

**S1 Text. Fig A. Source data for Fig 1. (A)** Sequence alignment of CG8223 with NASP homologs in other organisms. Darkening of the color indicates greater conservation. Magenta dots represent the α-N Histone H3 binding region observed in *Homo sapiens* NASP. Boxed region represents the gRNA target sequence for CRISPR-based mutagenesis to generate *NASP* mutants. **(B)** Representative image of DAPI stained 0-2hr AEL embryos with the percentage of stages 1–14 embryos from two replicates. **(C)** TMT abundance of each channel for mass spectrometry IP TMT normalized to total peptides. **(D)** Localization of NASP (red) in Drosophila S2 cells. DNA is stained by DAPI (blue). Scale bar, 5 μm. Graph displays intensity profiles of NASP and DAPI through a perpendicular line. **(E)** Localization of NASP (green) in Drosophila S2 cells. DNA replication is marked by CldU pulsing (red) and DNA is stained by DAPI (blue). Scale bar, 5 μm. Graph displays intensity profiles of NASP, CldU, and DAPI through a perpendicular line. **(F)** Western blot analysis of ovary extracts from the indicated genotypes with total protein loading control. Blot is cropped for Fig 2B. **Fig B. Source data for Fig 2. (A)** Number of embryos laid by wild type or *NASP²/Df(3R)Exel6150* mothers. Each data point is representative of a biological replicate from 70 females(n = 3). Unpaired t-test was used to determine significance ($p < 0.05$) **(B)** *DAFC-66D* copy number relative to a non-amplified control locus from stage 12 egg chambers for the genotypes listed on the x-axis. Kruskal-Wallis ANOVA was performed to determine significance ($p < 0.05$). **(C)** The number of pupae on day 10 produced from virgin females with the genotypes outlined on the x-axis crossed with wild type males. Each data point is representative of a biological replicate (n = 3). Unpaired t-test was used to determine significance ($p < 0.05$). **(D)** Percentage of embryos hatched laid by wild

type or *NASP²/Df(3R)Exel6150* mothers. Each data point is representative of a biological replicate (n = 4) and represents the hatch rate of a group of 100 embryos. Dunn's Multiple Comparison post-hoc was performed to determine significance (p<0.05). **Fig C. Source data for Fig 3**. **(A)** Western blot analysis of total, insoluble, and soluble protein preps from stage 14 egg chambers dissected from wild type or *NASP²/Df(3R)Exel6150* mutant mothers. **(B)**TMT abundance of each channel for mass spectrometry normalized to total peptides of 0-2hr embryos and stage 14 egg chambers laid or dissected from wild type or *NASP²/Df(3R)Exel6150* mothers. **Fig D. Source data for Fig 4**. **(A)** Representative single embryos cropped from max project images used to define nuclear cycle stages for scoring data presented in Fig 4A and 4B. Scale bar represents 100μm.

(PDF)

**S1 Data. Raw numerical data underlying graphs.**
(XLSX)

**S1 Table. Resource table for all reagents and software used in this manuscript.**
(DOCX)

## Acknowledgments

We thank Jacki Hao for assistance in generating the *NASP* mutants. We thank Amanda Amodeo and Andrea Page-McCaw for critical feedback on this manuscript.

## Author Contributions

**Conceptualization:** Reyhaneh Tirgar, Jared T. Nordman.

**Formal analysis:** Reyhaneh Tirgar, Jonathan P. Davies, Lars Plate.

**Funding acquisition:** Jared T. Nordman.

**Investigation:** Reyhaneh Tirgar, Jonathan P. Davies.

**Supervision:** Jared T. Nordman.

**Writing – original draft:** Reyhaneh Tirgar, Jared T. Nordman.

**Writing – review & editing:** Reyhaneh Tirgar, Jonathan P. Davies, Lars Plate, Jared T. Nordman.

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
