## [Editor Report · Decision Letter 0]

2 Dec 2022

Dear Dr Nordman,

Thank you very much for submitting your Research Article entitled 'The histone chaperone NASP maintains H3-H4 reservoirs in the early Drosophila embryo' to PLOS Genetics.

The manuscript was fully evaluated at the editorial level, including a review of the Review Commons reviews and your planned response. Based on the reviews, we will not be able to accept this version of the manuscript, but we would be willing to review a much-revised version. We cannot, of course, promise publication at that time. We are sending the manuscript back to you to complete the revisions you have proposed.

If you decide to revise the manuscript for further consideration at PLOS Genetics, please aim to resubmit within the next 60 days, unless it will take extra time to address the concerns of the reviewers, in which case we would appreciate an expected resubmission date by email to plosgenetics@plos.org.

We are sorry that we cannot be more positive about your manuscript at this stage. Please do not hesitate to contact us if you have any concerns or questions.

Yours sincerely,

Gregory P. Copenhaver, Ph.D.

Editor-in-Chief

PLOS Genetics

Gregory Barsh

Editor-in-Chief

PLOS Genetics

---

## [Decision Letter · Decision Letter 1]

15 Feb 2023

Dear Dr Nordman,

Thank you very much for submitting your Research Article entitled 'The histone chaperone NASP maintains H3-H4 reservoirs in the early Drosophila embryo' to PLOS Genetics.

The manuscript was fully evaluated at the editorial level and by three independent peer reviewers. The reviewers appreciated the attention to an important topic but identified some concerns that we ask you address in a revised manuscript (please note that Reviewer #1 indicated in their comments to the editor that they are satisfied with the revisions, so their "no" comment should be interpreted as "no more revision necessary"). You should be able to address all of the remaining comments by amending the text with the exception of Reviewer #2's suggestion that you present IF images of H3-H4 localization in stage 14 eggs.  I agree with the reviewer that IF images would strengthen the manuscript, and if you have that data or can easily obtain it I encourage you to add it, but eventual acceptance will not be contingent on it.  Once you make these revisions, I will be able to render a final editorial decision without further external review.

We therefore ask you to modify the manuscript according to the review recommendations. Your revisions should address the specific points made by each reviewer.

Yours sincerely,

Gregory P. Copenhaver

Editor-in-Chief

PLOS Genetics

Gregory Barsh

Editor-in-Chief

PLOS Genetics

Reviewer's Responses to Questions

**Comments to the Authors:**

Reviewer #1: no

Reviewer #2: Summary:

The rapid early embryonic cycles in Drosophila pose a need for maternal histone supply, as well as histone stabilisation by specific chaperones before deposition. This manuscript investigates the H3-H4 chaperone NASP homolog in Drosophila. Based on multiple lines of evidence - sequence homology, structural predictions, and biochemical co-purification with H3-H4 (by IP/MS) - the authors convincingly identify gene CG8223 as Drosophila NASP. Functional experiments show that soluble H3-H4 pools are destabilised in the absence of CG8223 and provide further support. NASP deletion by CRISPR effectively demonstrates that the null mutant is viable (using various deficiency lines). Moreover, NASP is a maternal effect gene and NASP mutant mothers have defects in embryonic development and embryo hatching. This study identifies the fly NASP homolog and effectively demonstrates its functional requirement in embryo development.

This revised version of the manuscript addresses most, if not all, of the previous concerns raised. I have three additional comments/suggestions:

1. The more in-depth characterisation of nuclear and segregation defects in early embryos now provided in Figure 4 strengthens the study. Also, the additional discussion of previous findings from Zhang et al 2018 provides clarity on the novelty of this current study. Related to this phenotype, the authors suggest the possibility in the Discussion that embryos laid by NASP mutant mother are not fertilised or develop as haploids. Is it not possible to already decipher this information from the images collected for quantitation in Figure 4?

2. With respect to NASP localisation, the authors were unable to confirm its localisation in embryos. They do however present results showing its localisation in cultured S2 cells, where it appears to be enriched in the cytoplasm. Some further discussion of this localisation pattern and its predicted localisation in the syncytium would be useful to explain/predict its function in embryos. Also, some further explanation of the observed localisation in S phase would be useful.

3. Based on Figure 3, the authors claim that H3 forms insoluble aggregates in stage 14 egg chambers and propose a specific role in solubilisation at this stage. IF images of H3-H4 localisation in stage 14 egg would significantly strengthen this claim. Also, there appears to be an error in the labelling of lanes in Fig S3A.

Minor:

Line 55: remove ‘provides’

Reviewer #3: The authors have addressed all my suggestions and I only have a few formal comments below (one is quite important, though). Congratulations on this fine work!

- These two sentences of the abstract are a bit redundant and could possibly be merged into one : « Here, we show that CG8223, the Drosophila homolog of NASP, is a H3-H4-specific chaperone in the early embryo. NASP specifically binds to H3-H4 in the early embryo. »

- In the sentence « Interestingly, we noticed a higher level of H3

and H3.3 in CG8223 IPs relative to H4, suggesting that CG8223 preferentially binds to H3 », I understand that H3 first designates H3.2 (the replicative Drosophila H3) and then is used as a generic name for H3. To avoid confusion, the authors could use H3.2 when it is relevant.

- Important point : in the discussion, the authors speculate on the possibility that defects observed in developing NASP embryos could result from a fertilization failure. This is actually not possible because fertilization is essential to initiate embryogenesis in Drosophila (the sperm provides centrioles to the egg). In embryos from Hira mutant females, eggs are fertilized but the male pronucleus is not properly formed. The corresponding paragraph should thus be rephrased.

**Have all data underlying the figures and results presented in the manuscript been provided?**

Reviewer #1: Yes

Reviewer #2: Yes

Reviewer #3: Yes

PLOS authors have the option to publish the peer review history of their article (what does this mean?). If published, this will include your full peer review and any attached files.

Reviewer #1: No

Reviewer #2: No

Reviewer #3: No

---

## [Editor Report · Decision Letter 2]

24 Feb 2023

Dear Dr Nordman,

We are pleased to inform you that your manuscript entitled "The histone chaperone NASP maintains H3-H4 reservoirs in the early Drosophila embryo" has been editorially accepted for publication in PLOS Genetics. Congratulations!

Yours sincerely,

Gregory P. Copenhaver

Editor-in-Chief

PLOS Genetics

Gregory Barsh

Editor-in-Chief

PLOS Genetics

Comments from the reviewers (if applicable):

**Data Deposition**

http://datadryad.org/submit?journalID=pgenetics&manu=PGENETICS-D-22-01308R2

**Press Queries**

---

## [Editor Report · Acceptance letter]

14 Mar 2023

PGENETICS-D-22-01308R2 

The histone chaperone NASP maintains H3-H4 reservoirs in the early Drosophila embryo 

Dear Dr Nordman, 

We are pleased to inform you that your manuscript entitled "The histone chaperone NASP maintains H3-H4 reservoirs in the early Drosophila embryo" has been formally accepted for publication in PLOS Genetics! Your manuscript is now with our production department and you will be notified of the publication date in due course.

With kind regards,

Zsofia Freund

PLOS Genetics

On behalf of:
